# PixelBot: Learning To Imitate Game Testing Behaviors from Offline Pixel-based Demonstrations

**Sherif Abdelfattah, Farrukh Rahman, Adrian Brown**

{sherifgad, farrukh.rahman, adrianbr}@microsoft.com

**Xbox Studios Quality AI Lab, Microsoft, Redmond, WA, USA**

## Abstract

Automated game inspection is increasingly crucial for maintaining the quality of complex 3D gaming environments. However, most current automation approaches are deterministic and require intrusive integration with the game engine. Artificial intelligence (AI) agents trained via imitation learning (IL) present a versatile alternative, as they can learn from quality engineer demonstrations. Despite this potential, deploying AI agents effectively in game inspection faces several obstacles. These challenges include the need for demonstration sample efficiency, the lack of explicit reward signals, restricted access to supplementary modalities or internal game data, and the critical demand for rapid inference speed. To address these issues, we propose an AI agent architecture named *PixelBot*. This architecture primarily utilizes pixel data (i.e., RGB frames) while maintaining sample efficiency for training with limited data. Our agent training methodology involves a two-stage process: first, a general approach for generating progress rewards from offline demonstrations, followed by return-modulated Behavioral Cloning (BC). We evaluated *PixelBot* across three Unreal Engine gaming environments, comparing its performance against established BC baselines. Our results demonstrate that *PixelBot* achieves an optimal balance between test imitation performance and parameter efficiency.

## 1 Introduction

The game industry has gained significant growth over the last decade Skwarczek (2021) as a result of advancements in 3D game engines, computing capacity, expansions in the Internet bandwidths, and the continuously increasing demand. Meanwhile, there was an increasing trend in game development complexity and scale to cope with such a growth rate. This demanded that game development studios use large teams of quality assurance engineers to guarantee the quality of their games. However, the increase in demand and game complexity poses a significant challenge in following this approach, as a larger workforce will translate to increased cost and management efforts. To cope with such a challenge, game development teams tend to rely on test automation procedures Politowski et al. (2022) to reduce the amount of manual labor work. While being effective, these automation procedures are usually executed through deterministic code or behavior trees Fronek et al. (2020) specifically tailored for the test scenario and using internal game engine integration modules. This deterministic and invasive nature limits the scaling of their adoption across multiple scenarios and gaming environments, and requires the involvement of game development engineers.

Artificial intelligence (AI) agents offer a promising solution to the scalability limitations of traditional test automation in game quality assurance by augmenting human involvement to automate various game-testing tasks. While diverse approaches exist for training such agents, including Reinforcement Learning (RL) Sutton & Barto (1998), Model Predictive Control (MPC) García et al.

(1989), and Imitation Learning (IL) Hussein et al. (2017), we contend that IL is the most pragmatic option in this domain due to its minimal assumptions regarding the deployment environment. These assumptions often include access to the internal game state for reward signal engineering, direct simulation within a resettable environment, or the necessity of learning a world model for planning. Typically, training an IL agent involves gathering demonstration samples from a human expert on task execution, followed by learning a policy designed to maximize the likelihood of replicating the expert's actions given similar observations. This methodology mirrors how humans learn from one another through direct demonstration. Within the context of game test automation, an optimal deployment workflow would involve a game quality engineer recording several demonstration trajectories of the test task, initiating IL training, and subsequently deploying and evaluating the trained agent's performance.

To ensure the effectiveness of such a workflow, several challenges must be addressed. First, the limited observation modalities in the demonstration dataset, primarily consisting of RGB data (e.g., screenshots), pose a significant hurdle. While incorporating additional observation modalities like depth or segmentation could enhance an agent's robustness and generalization, acquiring these modalities is costly and may necessitate integration with the game engine. Second, the inability to directly simulate against the game environment for agent fine-tuning presents another challenge. This limitation often requires invasive integration procedures to reset levels and bypass menus, a common requirement for reinforcement learning (RL) and adversarial imitation learning (IL) methods. Finally, the need for fast inference speed is crucial for real-time responsiveness in dynamic gaming scenarios, such as combat or racing. This challenge can lead to diminishing returns when employing complex architectures that perform well during training but poorly during deployment.

Recent approaches have explored training Imitation Learning (IL) agents within the gaming domain, demonstrating promising results in gameplay and exploration tasks Raad et al. (2024); Abdelfattah et al. (2023); Liu et al. (2022). However, a notable gap remains for approaches specifically tailored to game test automation and designed to overcome its inherent practical deployment challenges. Our proposed *PixelBot* agent addresses this gap through several key contributions. First, *PixelBot* learns directly from pixel-based observations, circumventing the requirement for additional modalities or access to internal game state information. Second, it offers a generic method for generating progress rewards from offline demonstrations and effectively fusing them with input features to guide the imitation policy. Finally, our efficient architecture design achieves a crucial balance between robust offline training and efficient deployment performance.

## 2 Problem Definition

Our proposed method aims at learning to imitate from expert demonstrations thus, we follow a supervised learning setup. Given an offline demonstration dataset $D = \{\tau^1, \tau^2, \ldots, \tau^N\}$, where $\tau^i = \{(a_1^p, s_1), (a_1^p, s_1), \ldots, (a_K^p, s_K)\}$ is a trajectory of state-action pairs with a length $K$, we optimize the parameters $\theta$ of an imitation policy $\hat{a}^p{}_t = \pi_\theta^i(s_t)$ by minimizing a cross-entropy loss $H$ with respect to the expert's policy $a_t^p = \pi^p(s_t)$:

$$\underset{\theta^* \in \theta}{\arg\min} \sum_{n=1}^{N} \sum_{(a_t^p, s_t) \in \tau^n} H(\pi_\theta^i(s_t), a_t^p) \tag{1}$$

## 3 Methodology

Our proposed agent design consists of three main modules, including 1) pixel observation feature extraction, 2) temporal dynamics modeling using recurrent memory, and 3) action and return prediction using specialized projection heads. We outline each of these modules as follows.

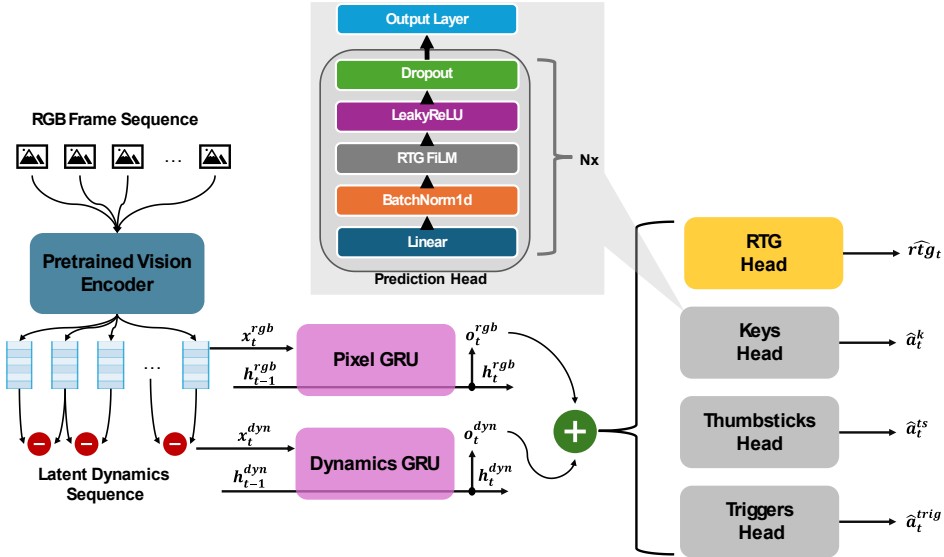

Figure 1: The PixelBot's model architecture. The model encodes the input pixel sequence into visual embedding and latent temporal dynamics streams. Then, GRU cells encode input streams to extract sequence flow. Finally, Xbox joystick action and return-to-go (RTG) are predicted using dedicated projection heads given the encoded sequence flow.

## 3.1  Feature Extraction from Pixel Data

Our agent mainly depends on pixel data (i.e., RGB frames) for representing vision observations. This design assumption minimizes dependency on internal game state information (e.g., player position, 3D semantic maps, game scores, etc.), as such dependency could be prohibitive in some application scenarios when the game development team lacks support capacity. We introduce inductive bias from large-scale image datasets to our vision feature extraction by using a pretrained vision encoder [1] to encode the input pixel sequence. We evaluated multiple pretrained vision encoders and found that the classic AlexNet Krizhevsky et al. (2012) encoder achieves a balance between performance and efficiency (see ablation results in Section 4 for more details).

## 3.2  Temporal Dynamics Modeling

We utilize a sliding temporal window of fixed size over pixel observations to formulate our state representation. Using action observations along with pixel ones is a common design choice for pixel-based imitation agents Raad et al. (2024), yet it also adds a significant amount of computation to encode and embed high-dimensional action spaces, such as a gamepad controller involving discrete keys and multiple analog signals (e.g., thumbsticks and triggers). To mitigate this challenge, we utilize deltas between pixel embeddings in the sliding window as a generic pseudo-action representation, which eliminates the need to input high-dimensional action data and makes our dynamic modeling agnostic to the action space (i.e., only action prediction heads will be dependent on the action space setup).

To model temporal dynamics from pixel and pseudo-action sequences, we propose a dual Gated Recurrent Units (GRUs) Chung et al. (2014) memory module (see the middle part of Figure 1). We also evaluated transformer-based variants Vaswani et al. (2017) and found that the dual GRU design achieves a balance between effectiveness in capturing dynamics features while being more parameter efficient. We present this evaluation in the ablation results in Section 4. GRUs are a simplified variant

---

[1] torchvision pretrained models

of Long Short-Term Memory (LSTM) Hochreiter & Schmidhuber (1997) designed to capture long-range dependencies in sequential data. They employ update ($z_t$) and reset ($r_t$) gates to control the flow of information, effectively managing memory and preventing vanishing gradients. The update gate determines how much of the previous hidden state is retained, while the reset gate controls how much past information is ignored. The candidate hidden state ($\tilde{h}_t$) is computed using the reset gate, and the final hidden state ($h_t$) is a weighted combination of the previous and candidate states, governed by the update gate.

The core equations defining the GRU's operation are:

$$z_t = \sigma(W_z x_t + U_z h_{t-1} + b_z)$$
$$r_t = \sigma(W_r x_t + U_r h_{t-1} + b_r)$$
$$\tilde{h}_t = \tanh(W_h x_t + U_h(r_t \odot h_{t-1}) + b_h)$$
$$h_t = (1 - z_t) \odot h_{t-1} + z_t \odot \tilde{h}_t$$

where $x_t$ is the input, $h_{t-1}$ is the previous hidden state, $\sigma$ is the sigmoid function, $\tanh$ is the hyperbolic tangent function, $\odot$ denotes element-wise multiplication, and $W$ and $b$ represent weight matrices and bias vectors, respectively.

Each cell within our dual GRU memory is dedicated to an input modality, including the pixel (Pixel GRU) and the pseudo-action (Dynamics GRU) sequences. The output of the two cells is concatenated and then used as input for the prediction heads. We found that this outperforms using one GRU cell working on an overlapped pixel and pseudo-action sequence (see Section 4 for ablation results).

### 3.3 Action and Return Prediction

We target an Xbox controller action space, which consists of discrete actions, including keypad buttons (around 15 unique values), and continuous actions, including left and right thumbsticks (value range $[-1, 1]$) and left and right triggers (value range $[0, 1]$). To relax the complexity of this action space, we transform the continuous thumbstick and trigger values into discrete buckets inspired by Foveated rendering Jabbireddy et al. (2022) to give more resolution to the middle values for fine control. To count for key combos (i.e., combinations of buttons pressed together), we predict it as a multi-binary target, where pressed buttons would have 1s and 0s for non-pressed ones. We use categorical cross-entropy loss for thumbstick and trigger buckets prediction (see Eq. 2), and multi-binary cross-entropy loss (see Eq. 3) for the keybad buttons prediction.

$$\mathcal{L}_{categorical\,CE} = -\frac{1}{N} \sum_{i=1}^{N} \sum_{j=1}^{C} y_{ij} \log p_{ij} \tag{2}$$

$$\mathcal{L}_{multibinary\,CE} = -\frac{1}{N} \sum_{i=1}^{N} \sum_{j=1}^{C} (y_{ij} \log p_{ij} + (1 - y_{ij}) \log(1 - p_{ij})) \tag{3}$$

To compensate for the absence of a reward signal and to furnish an additional guidance cue to our agent beyond pixel observations, we implemented a generic method for calculating progress rewards. These rewards are inversely proportional to the temporal distance between the current state and the terminal goal state within a given demonstration, normalized by the trajectory length. Subsequently, we transformed these progress rewards into a return-to-go (RTG) signal using a $\gamma$ discounting factor Sutton & Barto (1998).

$$rtg_t = \sum_{k=0}^{T-t} \gamma^k \cdot r_{t+k} \tag{4}$$

where the reward at time point $(t)$ is calculated as the progress till the end of the demonstration $r_t = \frac{t}{T}$.

Following Equation 4, we precompute return-to-go values in batch mode for the whole dataset. We modulate action prediction heads with RTG information using Feature-wise Linear Modulation (FiLM) Perez et al. (2018) layers. A FiLM layer works through two linear modulation gates (see Equation 5) that condition the prediction context of the GRU cells using the RTG estimate. We inject FiLM layers after batch normalization in each feedforward block as shown in Figure 1.

$$\mathbf{c}'_t = \gamma_c(\mathbf{rtg}) \cdot \mathbf{c}_t + \beta_c(\mathbf{rtg}) \tag{5}$$

where $c_t$ is the prediction context at time $(t)$, while $\gamma$ and $\beta$ are typically represented as linear functions $f(x) = Wx + b$.

The RTG prediction is handled by a dedicated head that follows a similar design as action prediction heads without the FiLM layers. We represent the RTG prediction as a regression objective predicting a single scalar value and use the Mean Squared Error (MSE) function (see Equation 6) to calculate its loss value. We note that during training time, we perform FiLM modulation using ground truth RTG values from the training dataset, while for evaluation and inference time, we utilize the RTG prediction head to predict the RTG, then we modulate action heads using the predicted RTG value.

$$\mathcal{L}_{\text{MSE}} = \frac{1}{N} \sum_{i=1}^{N} (y_i - \hat{y}_i)^2 \tag{6}$$

## 4 Experimental Analysis

This section delineates our experimental analysis, which investigates three principal research questions: **[RQ1]** What is the comparative performance of the proposed *PixelBot* agent against other Imitation Learning (IL) baselines across the chosen gaming environments? **[RQ2]** To what extent does our agent exhibit robustness when subjected to a low training data regime? And **[RQ3]** What is the individual impact of each *PixelBot* design choice on the holistic performance? The subsequent discourse details the experimental methodology employed to address these research questions.

### 4.1 Experimental Gaming Environments

We created three distinct gaming scenarios for our experiments, all built within the Lyra demo from Unreal Engine[2]. Each scenario is designed to simulate specific testing behaviors (e.g., testing weapons, ability to kill enemies, working sequence of buzzle/obstacles, etc.), aligning with the game's inherent logic and featuring unique visual characteristics. Figure 2 provides screenshots of these environments.

The top environment, which we call "Enemy Fighter", tasks the player with bypassing a wall obstacle and then pursuing and eliminating an evading enemy. The middle environment named "Obstacle Course", demands the player to bypass a sequence of obstacles such as opening doors by shooting on the green sphere lock, jumping over barriers, and crouching under obstacles to reach a trophy. Finally, the bottom environment named "Weapons Test", is focused on weapon testing by demanding the player to select a weapon (pistol, grenades, rifle, or shotgun) from a closet and then engage and eliminate multiple enemies with different colors.

For each experimental environment, we collect 50 trajectories (i.e., pixel observation and Xbox action pairs) demonstrating how to perform the task and split them with a $(90\% : 10\%)$ ratio for train and validation splits, respectively. We sampled our datasets at a 7 frames-per-second ratio. The validation split is used for hyperparameter fine-tuning.

---

[2]Unreal Lyra Demo

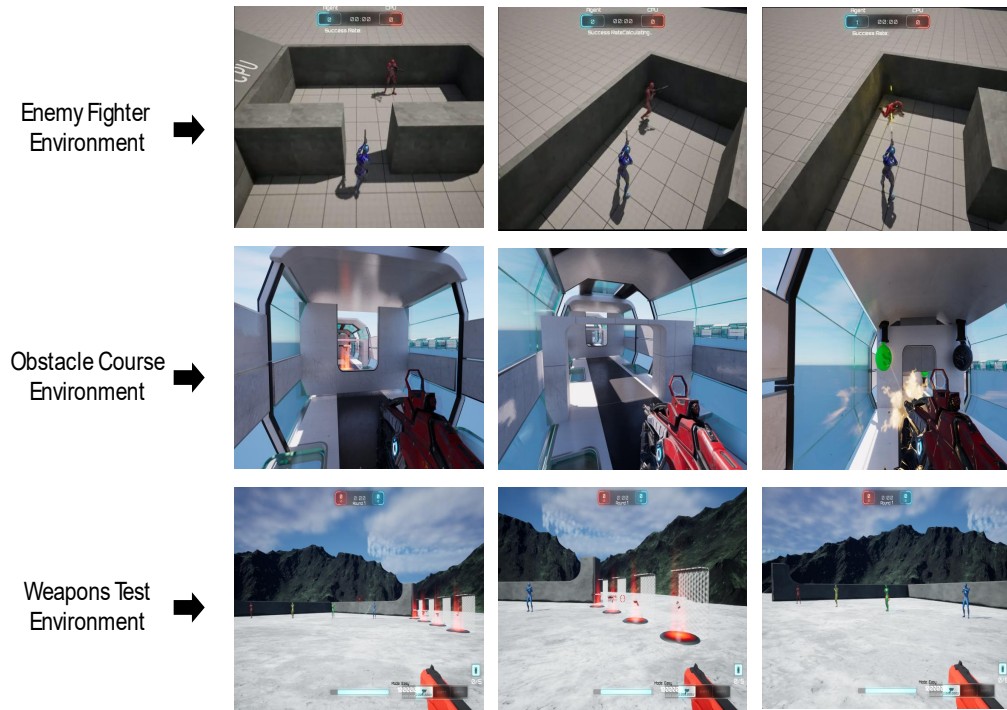

Figure 2: Experimental game environment. At the top, the enemy fighter environment. In the middle, the obstacle course environment, and at the bottom, the weapons test environment.

## 4.2 Comparison Baselines

We compared our *PixelBot* agent against three distinct baseline models. The first, a **Vanilla Behavior Cloning (VBC)** agent, concatenates encoded pixel data from the current sliding window. It employs Multi-Layer Perceptron (MLP) heads, structurally similar to our projection head but lacking FiLM layers, to predict Xbox actions. The second, a **Vanilla Recurrent (VR) agent**, utilizes a single Gated Recurrent Unit (GRU) cell. This agent processes an overlapping sequence of pixel and pseudo-action inputs from the current sliding window and uses MLP heads (also without FiLM layers) to predict Xbox actions. Finally, a **Decision Transformer (DT) agent** Chen et al. (2021) predicts Xbox actions using triplets of pixel observations, actions, and return-to-go values obtained from the current sliding window.

## 4.3 Performance Evaluation Compared to Baselines

To answer research questions **RQ1** and **RQ2**, we conducted two distinct training cycles: one utilizing 100% of the training dataset and another employing 50% of the training dataset. This configuration was designed to rigorously evaluate the robustness of each comparison model when operating under a low training data regime, and to systematically quantify the resultant performance degradation. For each training cycle, we train for 50 epochs. Following this, we conducted 10 independent evaluation runs within each experimental environment. The success rate was calculated as the primary performance metric.

Figure 3 illustrates the success rate achieved by each agent across the three experimental environments with the two training cycles. The left bars represent performance after training with 50% of the dataset, while the right bars illustrate performance with 100% of the dataset. We observe that the *PixelBot* agent outperforms all the other baselines with margins of 9.2%, 4.9%, and 4.7% from

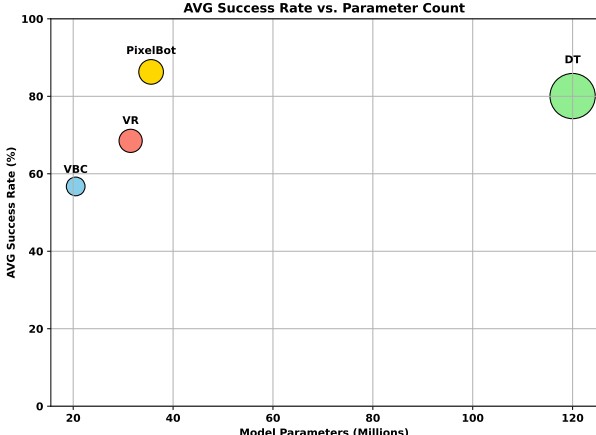

Figure 3: Success rate comparison results for each agent over the three experimental environments.

Figure 4: Comparing AVG success rate vs. total number of parameters for each agent.

the nearest performer (the DT agent) when training with $100\%$ of the training dataset. Our model additionally maintained superior performance when utilizing $50\%$ of the training dataset, exhibiting a minimal average success rate degradation of $-2.6\%$ across all scenarios. This contrasts with a $-10.8\%$ degradation observed in the DT agent, which was the closest top performer. We posit that the more substantial decline in the DT model's performance is attributable to overfitting, likely due to its extensive parameter space. We observed that the success rate in the "Obstacle Course" scenario was generally higher compared to the other ones. This could be attributed to a lower probability of action noise during deployment in this scenario, as it primarily involves a linear sequence of obstacles. In contrast, the other environments likely experienced higher noise due to factors such as moving enemies or the presence of compounding tasks, such as the prerequisite of acquiring weapons prior to engaging adversaries.

To further elucidate the potential of the *PixelBot* agent, we computed each agent's average success rate across all experimental environments. This metric was subsequently plotted against the total number of parameters for each agent, allowing for an assessment of their respective trade-offs between effectiveness and efficiency. The results are presented in Figure 4. Notably, despite the *PixelBot* agent possessing approximately one-third ($\sim 35.6$M) of the total number of parameters of the DT agent ($\sim 120$M), it consistently outperformed the DT agent across all experimental environments, thereby achieving the most favorable balance between performance and model parameter count.

### 4.4 Ablation Study

This section addresses research question **RQ3** by evaluating the impact of key design choices within our proposed *PixelBot* agent. Specifically, we investigate the influence of vision encoder selection, the length of the context window, and the integration of RTG FiLM modulation. We compare the average success rate of the optimal *PixelBot* configuration against each ablated variant. Table 1 (see Appendix A) summarizes the findings of our ablation study across these distinct design choices.

The impact of RTG modulation is evident in the "No RTG FiLM" variant, as its removal resulted in a 3.5% decrease in the average success rate. Interestingly, we found that employing more complex encoders, such as ResNets He et al. (2016) and EfficientNet Tan & Le (2021), did not yield superior performance compared to AlexNet Krizhevsky et al. (2012). This could be attributed to AlexNet's wide CNN design, which has demonstrated greater effectiveness in certain agent learning setups Espeholt et al. (2018). Finally, we observed that extending the temporal length of the context window from 1 second to 2 and 3 seconds did not result in a better success rate. This could be due to the added inference latency required to process longer context data, which diminishes the expected returns from expanding the context window length.

## 5  Related Work

Recently, several AI-driven methods have emerged to automate game testing. Chang et al. introduced Reveal-More Chang et al. (2019), an exploration technique that enhances human testers' efforts in increasing game coverage. Their method involves an initial manual exploration by a human to pinpoint effective states for trials, followed by a Rapidly Exploring Random Trees (RRT) algorithm LaValle & Jr. (1999) that conducts random exploration walks from these identified states. Zheng et al. Zheng et al. (2019) framed game testing as a multi-objective optimization challenge, balancing exploration and gameplay. Their approach combines reinforcement learning (RL) with evolutionary selection to iteratively develop a set of non-dominated policies that optimize both objectives. Similarly, Bergdahl et al. Bergdahl et al. (2020) used deep RL to simultaneously train multiple goal-seeking agents, enabling them to uncover unusual paths by using the distance to a goal as a reward function.

While effective, the aforementioned RL-based approaches heavily depend on internal game state representations, such as global positioning, semantic maps, and game scores. This reliance restricts their generalizability across diverse games and necessitates invasive integration through developer-provided APIs. In contrast, Imitation Learning (IL)-based agents that can learn directly from pixel observations Raad et al. (2024); Abdelfattah et al. (2023); Liu et al. (2022) offer a more practical alternative for automating game testing. Our proposed *PixelBot* agent adopts this latter approach.

## 6  Conclusion & Future Work

In this work, we introduced *PixelBot*, an AI agent engineered with several key characteristics to address the practical deployment challenges inherent in game test automation. These features include its primary reliance on pixel observations, obviating the need to stack action observations in the input, and its provision of a generic method for generating and fusing progress rewards from offline demonstrations. Comparative results across three distinct gaming scenarios, evaluated against three imitation learning baselines, consistently underscored *PixelBot*'s effectiveness and its favorable balance of parameter efficiency when compared to the next best-performing baseline.

In future work, we plan to investigate pretraining methods, such as learning world models from mixed video and demonstration datasets, to further enhance generalization and accelerate transfer across different deployment scenarios. Another interesting avenue for future investigation is to explore extending our agent design to support hierarchical control over atomic skills. Finally, we aim to evaluate *PixelBot* on more complex gaming titles.

# A   Ablation Results

We summarize the ablation results in Table 1 as follows.

Table 1: Ablation study for different design choices in the PixelBot agent.

| PixelBot Variant | AVG Success Rate |
| --- | --- |
| PixelBot | **86.3**% |
| PixelBot (2 second window) | 85.8% |
| PixelBot (3 second window) | 84.7% |
| PixelBot (No RTG FiLM) | 82.8% |
| PixelBot (ResNet-50) | 79.4% |
| PixelBot (ResNet-18) | 76.7% |
| PixelBot (EfficientNetV2) | 78.6% |

# B   Implementation Details

Table 2 summarizes the parameter configuration of our PixelBot architecture.

Table 2: PixelBot Agent Parameter Configuration

| Parameter | Configuration |
| --- | --- |
| Vision Encoder | pretrained AlexNet |
| Pixel observation resolution | $224 \times 224$ |
| Pixel augmentations | color jitter |
| GRU hidden state dim | 2048 |
| GRU hidden layers | 3 |
| GRU dropout | 0.1 |
| Feedforward blocks per prediction head | 2 |
| Feedforward block linear dim | 1024 |
| Feedforward block dropout | 0.2 |
| Temporal sliding window size | 1 second |
| Xbox action thumbstick bins | 21 |
| Xbox action trigger bins | 10 |
| RTG $\gamma$ | 0.99 |
| Optimizer | AdamW [lr:$1e^-4$, decay:0.1, beta1:0.9, beta2:0.95] |

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
