# OpenReview forum: "PixelBot: Learning To Imitate Game Testing Behaviors from Offline Pixel-based Demonstrations"
_rl-conference.cc/RLC/2025/Workshop/RLVG — RLVG Workshop - RLC 2025_

### Official Review · Reviewer_ERPQ · 2025-06-14
**This paper describes PixelBot, a game testing bot that learns from pixels and player data.**

**Rating:** 3
**Confidence:** 4

**Summary:**

The paper introduces PixelBot, a pixel-based testing agent designed to learn game testing through demonstrations, surpassing existing Behavior Cloning (BC) methods. Its architecture includes three modules: extracting pixel observation features, modeling temporal dynamics with recurrent memory, and predicting actions and returns via specialized projection heads. While innovative and promising as a testing bot, PixelBot's evaluation is limited to comparisons with BC methods and tested only on three levels of a single game. Additionally, its reliance on a computationally expensive vision model and player data poses challenges. The paper suggests broadening comparisons to Offline RL and other imitation learning methods, and testing in diverse environments beyond FPS games.

**Strengths:**

A novel approach that has merit as a testing bot.

**Weaknesses:**

- Only compares to other BC methods.
- Tested on only 3 levels in one game. The authors acknowledge that which is good.
- Uses a vision model, which is computationally expensive and sometimes prohibited in testing (impacts performance). For further details why this might not be desirable please consult previous paper explaining this: Gillberg et. al: Technical Challenges of Deploying Reinforcement
Learning Agents for Game Testing in AAA Games https://arxiv.org/pdf/2307.11105
- This approach needs player data to train. This is not always the case that this exists or that it can be retrieved.

**Best Paper Nomination:**

No

**Claims:**

- A new method to testing bots that uses a pre-trained vision model.

The experiments supports these claims albeit not convincing for different games or even comparing to other imitation learning methods that are not BC.

**Suggestions:**

- Compare with Offline RL and other imitation learning regimes.
- Verify results on another environment that is not a FPS.

---

### Official Review · Reviewer_vvmo · 2025-06-16
**PixelBot: Learning To Imitate Game Testing Behaviors from Offline Pixel-based Demonstrations**

**Rating:** 3
**Confidence:** 4

**Summary:**

The authors introduce a new method that more efficiently performs behaviour cloning from image input only. It is motivated that the agents trained with imitation learning can help automate game testing in complex 3d gaming environments. The proposed method shows that by predicting a reward value derived from the offline demonstrations with a custom architecture named PixelBot, the method behaves better than the baselines with fewer parameters.

**Strengths:**

Well written

The introduced agent offers a significant improvement over the baselines

The ablation section

**Weaknesses:**

Film layers are not defined in the paper; it would be useful to be there, since it is part of the main contribution. Without this explanation is hard to understand what is the impact of predicting the RTG in the Film Layers. I feel this is more important than the GRU equations, as GRU is a known architecture.

The baselines have different number of parameters. For example, VBC and VR have fewer parameters than PixelBot, so the comparison may not be entirely fair.

**Best Paper Nomination:**

No

**Claims:**

Yes

**Suggestions:**

It would be cool if the authors open-sourced the code, and the repo had a simple-to-use interface/guide to create the human demonstrations and how to train the agent to test games. I think it might also make a stronger paper.

---

### Decision · Program_Chairs · 2025-06-19

**Decision:**

Accept

**Comment:**

This paper introduces **PixelBot**, a novel method for more efficiently performing behavior cloning from image input to automate game testing in complex 3D gaming environments, demonstrating improved performance over baselines with fewer parameters by predicting a reward value from offline demonstrations.

The paper's strengths include its well-written presentation, the significant performance improvement PixelBot offers over baselines, and a thorough ablation section.

However, reviewers mentioned there was a lack of a clear definition for "Film layers," a potentially unfair comparison due to differing parameter counts in baselines, and a limited evaluation confined to only behavior cloning methods on three levels of a single game. We encourage the authors to address these points by defining all novel architectural components and adding a discussion of the fairness of baseline comparisons in the camera-ready version for presentation at the workshop.